# Metallurgical and Mechanical Characterization of High-Speed Friction Stir Welded AA 6082-T6 Aluminum Alloy

**DOI:** 10.3390/ma12244211

**Published:** 2019-12-14

**Authors:** Anton Naumov, Iuliia Morozova, Evgenii Rylkov, Aleksei Obrosov, Fedor Isupov, Vesselin Michailov, Andrey Rudskoy

**Affiliations:** 1Peter the Great St. Petersburg Polytechnic University, 195251 St. Petersburg, Russian; rylkov.en@edu.spbstu.ru (E.R.); isupov_fyu@spbstu.ru (F.I.); rector@spbstu.ru (A.R.); 2Department of Joining and Welding Technologies, Brandenburg University of Technology Cottbus-Senftenberg, 03046 Cottbus, Germany; michailov@b-tu.de; 3Department of Metallurgy and Materials Technology, Brandenburg University of Technology Cottbus-Senftenberg, 03046 Cottbus, Germany; Aleksei.Obrosov@b-tu.de

**Keywords:** high-speed friction stir welding (HSFSW), aluminum alloy, remnant oxide line (ROL), microstructural characteristics, mechanical properties, fracture behavior, FEM

## Abstract

The objective of this study was to investigate the effect of the high welding speed on the mechanical properties and their relations to microstructural characteristics of butt friction stir welded joints with the use of 6082-T6 aluminum alloy. The aluminum sheets of 2.0 mm thick were friction stir welded at low (conventional FSW) and high welding speeds (HSFSW) of 200 and 2500 mm/min, respectively. The grain size in the nugget zone (NZ) was decreased; the width of the softened region was narrowed down as well as the lowest microhardness value located in the heat-affected zone (HAZ) was enhanced by HSFSW. The increasing welding speed resulted in the higher ultimate tensile strength and lower elongation, but it had a slight influence on the yield strength. The differences in mechanical properties were explained by analysis of microstructural changes and tensile fracture surfaces of the welded joints, supported by the results of the numerical simulation of the temperature distribution and material flow. The fracture of the conventional FSW joint occurred in the HAZ, the weakest weld region, while all HSFSW joints raptured in the NZ. This demonstrated that both structural characteristics and microhardness distribution influenced the actual fracture locations.

## 1. Introduction

Friction stir welding (FSW) is recognized as one of the most efficient methods for joining of aluminum alloys, as it enables to avoid defects associated with fusion welding processes [1,2]. Although FSW has been widely used to weld the heat-treatable aluminum alloys of 6xxx series (Al-Mg-Si), the softening problem in the weld region, especially in the ultra-thin sheets, has not been solved yet. It is well-known that dispersion hardening of the heat-treatable aluminum alloys is based on the precipitation of the metastable precursors of the equilibrium β-Mg_2_Si particles. The mechanical properties of these friction stir welded alloys are mainly determined by size, volume fraction, and distribution of the precipitates in different weld zones. Various methods such as adjustment of the welding and tool parameters, ultrasonic vibrations, applying of the backing plate, and post-weld heat treatment cannot reach the mechanical properties of the base metal [3,4,5,6,7,8]. Numerous FSW studies of the 6xxx series alloys [9,10,11,12,13,14,15] have shown that the processes of dissolution and growth of the strengthening precipitates in the nugget zone (NZ), thermo-mechanically affected zone (TMAZ), and heat-affected zone (HAZ) lead to the softening in these zones. Together they form a softened region. According to the previous investigations, precipitation dissolution and coarsening are strongly controlled by the thermal cycle during welding, which in turn depends on the welding speed and tool rotational speed. Moreover, the welding speed is one of the key factors, which determines the maximum temperature of the welding process, and the cooling rate [16,17]. With increasing welding speed (at constant rotation speed and axial force), the time interval of the thermal cycle at elevated temperature is shorter. As a result, more precipitates remain in the solid solution before precipitation reaction begins. Therefore, a carefully controlled thermal cycle during welding allows us to manage the final weld microstructure and mechanical characteristics of the produced joint.

One of the factors hindering the widespread application of FSW is its relatively slow welding speed, which is usually hundreds of millimeters per minute. Recent developments in tool and equipment for the FSW process have enabled an implementation of the higher welding speeds up to 3 m/min [18]. Previous research findings of the high-speed friction stir welding (HSFSW) [19,20,21,22,23,24,25,26] have demonstrated that the defect-free joints of various aluminum alloys (even by welding of ultra-thin sheets) can be observed after HSFSW through an adjustment of the welding parameters and tool geometry. In [20,21] it has been reported that increasing welding speed results in the formation of a narrower HAZ and hardness increase in the HAZ and NZ. An improvement of the hardness in the HAZ of the AA 6082-T6 alloy welded at high welding speed has a positive influence on the weld yield strength efficiency [20]. On the contrary, Liu et al. [22] concluded that the thermal cycle implemented at HSFSW of AA6061-T6 aluminum sheets by high rotation speed led to the expansion of a softened zone that caused the deterioration of the mechanical properties. The results obtained in [23] revealed that the joints produced at the maximum tool rotational speed and minimum welding speed (called «hot» welds) possessed higher mechanical properties compared to the joints welded at low rotation speed and high welding speed (called «cold» welds). Sheikhi et al. found that lower heat input during welding affected by the welding speed or tool rotational speed led to the formation of finer grain structure in the NZ. However, it had no influence on the mechanical properties of the friction stir welded joints, but influenced the location of the tensile fracture [25].

The studies presented above provide evidence that the welding speed has an observable impact on the weld structure, but its influence on the mechanical behavior of the welded joints has not been sufficiently studied. In this work, the weld joints of AA 6082-T6 aluminum sheets after high-speed friction stir welding were investigated. The effect of welding speed on the weld structure and mechanical behavior was analyzed through characterization of the microstructure evolution, fractographic investigation of the tensile surfaces of the conventional and high-speed FSW joints, supported by the results of the numerical simulation of temperature distribution and material flow.

## 2. Materials and Methods

### 2.1. Friction Stir Welding

In this study the heat treatable aluminum alloy 6082-Т6 with nominal composition in wt% 0.76Si-0.53Mg-0.22Mn supplied in sheets 2 mm thickness has been used. The sheets were cut to the dimensions of 400 mm in length and 100 mm in width, respectively. Conventional and high-speed friction stir welds were performed parallel to the rolling direction of the plates by means of a five-axis FSW machine Matec 40P (Matec GmbH, Kongen, Germany), using different process and tool parameters, as presented in Table 1. The process and tool parameters for the conventional FSW were selected based on the results of the previous research to provide defect free welds [27,28,29,30,31]. For the HSFSW, the process parameters, which were chosen using the bibliographic references [20,21,22] as well as results of the previous welding tests, were used. During these welding trials, the rotation and welding speeds were varied in the ranges of 2000–2400 rpm and 2200–2600 mm/min respectively, with the step of 100 rpm or mm/min at the constant axial force of 6 kN and tilt angle of 1.2°. After welding, all the welded plates were visually examined for surface defects like excessive flash, sheet thinning, and surface flaws. According to the observation, the optimal parameters for HSFSW were selected for the further investigation (Table 1). The tool used for the performing of conventional FSW is composed of a smooth concave shoulder with a diameter of 12 mm and a smooth cylindrical probe with a diameter and length of 4 and 1.8 mm, respectively. The HSFSW was employed using the tool consisted of a smooth shoulder with a diameter of 10 mm and the conical probe which was tapered from 3 mm at root diameter to 2 mm at tip diameter with a length of 1.75 mm. The tilt angle (α) was varied for different FSW types, as shown in Table 1. 

As mentioned in the literature review, the heat input during welding is an important factor, since it determines the microstructure evolution in the weld zones causing changes in the global mechanical properties of the welded joint. Since the process parameters used by the studied FSW processes were various, it was necessary to calculate different relative values in order to estimate influence of the process parameters on the metallurgical and mechanical behavior of the FSW joints. The weld peach WP (mm/rotation) factor was determined as a relation between rotation speed N (rpm) and welding speed v (mm/min), Table 1.

In recent study the total heat input Q_total_ (Watt) has been calculated using a contact shear stress τ_contact_ (Pa), tool angular rotation speed ω (rad/s), a shoulder radius R_shoulder_ (mm), a probe radius R_probe_ (mm), and length R_probe_ (mm) by Equation (1):Q_total_ = 2/3π τ_contact_ ω (R^3^_shoulder_ + 3R^2^_probe_H_probe_).(1)

The contact shear stress τ_contact_ and tool angular rotation speed ω were determined with a friction coefficient µ = 0.4, an axial force F_z_ (N), a rotation speed N (s^−1^) using Equations (2) and (3), respectively:τ_contact_ = µF_z_/πR^2^_shoulder_.(2)
ω = 2πN/60.(3)

The linear energy l (J/mm) has been calculated as a relation between the heat input and welding speed v (mm/min). The significant difference in the heat input as well as linear energy of the conventional and high-speed FSW processes is highlighted in Table 1. The heat input by HSFSW was higher, while the linear energy was almost eight times lower compared to the conventional FSW.

### 2.2. Microstructural and Mechanical Properties Analysis

Samples for optical microscopy and hardness testing were cut perpendicular to the welding direction in accordance with Figure 1 and prepared by standard metallographic technique. The polished samples were etched with Weck reagent (1 g NaOH, 4 g KMnO4, and 100 mL H_2_O) for 10 s. The Vickers microhardness was measured along the centerline of the transverse weld section with a load of 0.980 N (HV 0.1) for 10 s after several weeks. The grain size was estimated by the linear intercept method. The characterization of the coarse particles was performed using the scanning electron microscope (SEM) TESCAN MIRA II (Brno, Czech Republic). The detection limit for the determination of their size was 0.5 µm.

Tensile tests were carried out on a tensile machine Walter + bai AG (Löhningen, Switzerland) using 10 mm/min crosshead speed at room temperature. The trials were performed up to the rupture of the tested specimens. The mechanical properties of the weld (ultimate tensile strength, yield stress, and elongation) were determined by testing ten tensile specimens of each weld. The tested specimens were positioned perpendicular to the welding direction, Figure 1. Fractographic analysis of the tensile fracture surfaces was performed directly after tests by SEM equipped with a detector for energy-dispersive X-ray spectroscopy (EDX). 

### 2.3. Numerical Simulation

Computational fluid dynamics (CFD) model was used to predict the temperature and effective strain rate distribution during FSW. The computational domain of the FSW model, tool position, and tool geometry are presented in Figure 2.

It should be noted that the tool was in contact with the workpiece only part of the shoulder surface. Material flows into the computational domain through the ‘Inlet’ and ‘Outlet’ boundaries with 200 and 2500 mm/min speeds, which were equal to the welding speeds applied in the experiments. The speeds of the ‘Sides’, ‘Top surface’, and ‘Down surface’ were also set as 200 and 2500 mm/min. The convection heat transfer coefficient was assumed to be 300 W/m^2^K at the ‘Down surface’ and 30 W/m^2^K at other surfaces. The surface that contacts with the tool was rotated around the Y axis with the speed according to the used parameters (Table 1). The CFD model contains 8637083 tetrahedral cells with finer mesh in the center. 

The frictional boundary condition proposed in [32,33] was applied in the presented study. The frictional stress at the interface may be represented as: [34]
(4)$$\vec{\tau }=\{\begin{array}{c} \mu_{f}\sigma_{n}\frac{{\vec{\upsilon }}_{vel}}{{\vec{\upsilon }}_{vel}}\tanh(\alpha {\vec{\upsilon }}_{vel}),\;\;if\;\;{\vec{\upsilon }}_{vel}\neq 0\\ 0,\;\;if\;\;{\vec{\upsilon }}_{vel}=0 \end{array},$$
where ${\vec{\upsilon }}_{vel}$ is the relative velocity between the tool and material; α = 20 s/m is a scaling constant; μ_f_ is the sliding friction coefficient; and σ_n_ is the normal pressure at the welding tool/workpiece interface. The normal pressure was calculated at each cell face at tool/workpiece interface. The dependence of the friction coefficient on temperature is unknown, but it is known that the value of the friction coefficient decreases with increasing temperature. The friction coefficient was taken equal to 0.4, and it linearly decreases after 350 to 0 °C at the solidus temperature. The interfacial friction heat is given by:(5)$$q_{f}=\eta \left\|{\vec{\tau }}_{f}\right\|\left\|{\vec{\upsilon }}_{vel}\right\|.$$

Here ${\vec{\tau }}_{f}$ is the frictional tangential force; ${\vec{\upsilon }}_{vel}$ is the relative velocity between the welding tool and the workpiece; and η = 0.7 is the ratio of the heat absorbed by the workpiece [32]. Plastic deformation heat input was considered as:(6)$$q_{p}=K\sigma \dot{\varepsilon },$$
where K = 0.6 is the mechanical efficiency; σ is the flow stress; and $\dot{\varepsilon }$ is the effective strain rate.

Flow stress of the workpiece was considered to be dependent on temperature and strain rate. It can be obtained by: [33]
(7)$$\sigma_{f}=Ae^{m_{1}T}\varepsilon^{m_{2}}e^{m_{4}/\varepsilon }{\left(1+\varepsilon \right)}^{m_{5}T}e^{m_{7}\varepsilon }{\dot{\varepsilon }}^{m_{8}T}.$$

Here σ_F_ is flow stress, ε is a strain, A is a constant, m_1_ to m_8_ are exponents expressing the influence of the deformation conditions on the stress, T (°C) is the deformation temperature, and $\dot{\varepsilon }$ is an effective strain rate. In the presented work, ε = 1.2, as a maximum value in the range of validity [35]. Effective strain rate was determined by:(8)$$\dot{\varepsilon }=\;\sqrt{\frac{2}{3}{\dot{\varepsilon }}_{ij}{\dot{\varepsilon }}_{ij}},$$
where ${\dot{\varepsilon }}_{ij}$ is the component of strain rate tensor, defined as:(9)$${\dot{\varepsilon }}_{ij}=\;\frac{1}{2}\left(\frac{dv_{i}}{dx_{j}}+\frac{dv_{j}}{dx_{i}}\right).$$

Since there is an error between the calculated and experimental results at a temperature close to the solidus, the flow stress at high temperature was modified by the method proposed in [36]. The viscosity of the material during the simulation of FSW is given by: [36,37]
(10)$$\mu =\frac{\sigma }{3\dot{\varepsilon }}.$$

## 3. Results

### 3.1. Macrostructural Characteristics 

The surface defects were not detected by visual examination, but the internal defects were revealed using metallographic analysis. Figure 3 shows the typical cross-sectional macrostructures of the conventional and high-speed friction stir welds which were studied by employing optical microscopy. The advancing side (AS) and retreating side (RS) of both welds were marked on the right and left side, respectively. The continuous thin zigzag line extending in the weld center from the top surface to the bottom was revealed in the etched macrograph of the conventional FSW joint, Figure 3a. This appeared to be a remnant oxide line (ROL), which represents the oxide films (e.g., Al_2_O_3_) from the initial butt surfaces broken up by tool during the stirring process [38,39,40]. The ROL was also observed in the HSFSW joint, however, it had different features. The joint line remnant passed through the whole nugget zone from top to bottom and possessed a ‘broken zigzag’ pattern, as shown in Figure 3b, which indicated the different material flow during HSFSW process. Moreover, it was found that the oxide line was denser and more concentrated in the case of HSFSW joint, and microcracks even formed along the line.

In the macrostructures of the studied welds three zones can be traditionally distinguished: the nugget zone (NZ) located in the weld center and characterized by a fine-grained recrystallized microstructure; the thermo-mechanically affected zone (TMAZ), which was composed of highly deformed grains; the heat-affected zone (HAZ), where the structure was influenced only by temperature; and the grain structure was almost the same as the structure of the unaffected base metal (BM). In the macrostructures the squares for the detailed microstructure examination in the mentioned zones are depicted in Figure 3 and are discussed in Section 3.2. As can be seen from the images of the cross macrosections, the geometry of the zones has been changed as a result of different welding and tool parameters. The nugget zone of the HSFSW joint acquired a more ‘localized’ form and lower width due to both the decreased linear energy and the tool of different geometry. 

### 3.2. Microstructural Characteristics

The optical microstructure as well as SEM micrograph of the BM are presented in Figure 4. The average grain sizes of the slightly elongated grains were 38 µm long and 20 µm across the rolling direction. The parent metal microstructure also contained the coarse particles of the second phase in the size range from 1 to 5 µm, elongated in the rolling direction and mainly distributed inside the grains. The composition of the second phase particles was examined by the qualitative EDX analysis. The results are given in Table 2 (Spectrum 1). The particles were rich in Al, Fe, Mn, Si, and Cu, suggesting the presence of complex intermetallic compounds type AlFe(Mn)Si. The observation of such intermetallic particles in the structure of 6xxx aluminum alloys was reported in other investigations [12,41,42].

The microstructure of the base metal was sufficiently affected by substantial plastic stirring during FSW process. Figure 5 demonstrates the microstructures in different zones of the studied FSW joints. The dissimilarity in the microstructures of the TMAZ and NZ obtained by different welding speeds was significant. The transition zone, which distinguishes the microstructure in the NZ from the HAZ, was more drastic in the case of the HSFSW, especially on the RS. The grains in the TMAZ were severely deformed and inclined to the plate surface by high strain. As a result, the border between the NZ and HAZ of the HSFSW joint were very abrupt, Figure 5d. In contrast, it was difficult to observe deformed grains in the structure of the conventional FSW joint. The boundaries of the NZ on the AS of both conventional and high-speed welds were fuzzier compared to the RS, so that the transition line was much more gradual, Figure 5c,f.

The microstructures in the NZ of the studied welds appeared as very fine equiaxed grains resulting from recrystallization process, but the difference in grain size and distribution was observed for different welding parameters The grains in the NZ of the conventional FSW joint varied in the wide size range from 1 to 16 µm; the average grain size was about 6.1 µm. The average size of the equiaxed grains in the NZ of the high speed weld was about 3.8 µm. This indicates that the increase of welding speed leads to significant grain refinement; thereby the nugget microstructure appeared to be finer and demonstrated a homogeneous distribution in contrast to the high gradient in grain size in the conventional NZ.

In the microstructures of the NZ of both studied welds the second phase particles were also observed, Figure 6. They were uniformly distributed predominantly along the grain boundaries. The obtained chemical compositions of these inclusions confirmed the evidence of the AlFe(Mn)Si intermetallics detected in the BM microstructure, as can be seen from Table 2 (Spectrum 2 and 3). These results allow us to suggest that the intermetallic particles did not dissolve during FSW due to the high melting point, and their intergranular location indicated redistribution by the stirring process. High thermodynamical stability of the second phase particles during FSW, as well as the possibility to be broken in the NZ were also noted in previous studies [12,41,42,43]. The average size measured on a large number of the particles in the NZ was about 2.5 and 2 µm for the conventional and HSFSW joints, respectively. A slight decrease in particle size in the NZ of both welds in comparison with BM can be explained by the fragmentation of the second phase particles by the stirring process.

### 3.3. Microhardness and Tensile Behaviour 

The results of the microhardness measurements are summarized in Figure 7. The microhardness of the base metal was about 116 HV. The microhardness of both welds had a W-shaped profile with a wide softened HAZ region that is typical for the FSW-welds made of heat treatable aluminum alloys. The hardness profile of the conventional FSW joint was almost symmetrical with respect to the weld center while the profile of the hardness distribution across conventional FSW joint was significantly shifted to the AS. The minimum hardness values of the joint produced at lower welding speed were 68 HV and 71 HV located on the RS and AS, respectively. The corresponding values in the case of the HSFSW were 77 HV on the RS and minimum microhardness point 69 HV on the AS. As can be seen, the softened region of the conventional joint was much wider and overall softening was significantly higher compared to weld produced at high welding speed. The strengthening in the HAZ promoted by the increase of the welding speed could be especially noted on the RS. Remarkably, the studied welds exhibited almost the same microhardness in the NZ. However, the average microhardness in the NZ of the HSFSW joint was slightly higher and inhomogeneous compared to the conventional weld, and it was found to be around 85 HV.

The mechanical properties of the studied joints were evaluated trough tensile testing. Figure 8 displays the tensile properties of the studied welds and base material, as well as pictures of the welded specimens after failure. There is a clear trend of decreasing mechanical properties by FSW process. The increase in the welding speed improved the tensile strength; however, it had little influence on the yield strength and significantly reduced elongation. During the tensile testing, the mechanical properties of the weakest section are accepted to be the global properties of the joint. Traditionally, the fracture of the heat-treatable aluminum alloys after FSW takes place in the HAZ region considering an absence of defects. As described in the scientific literature, the location of the weakest region is explained by the dissolution and coarsening of the strengthening precipitates due to the thermal influence, resulting in a decrease in mechanical properties. It can be highlighted that the tensile specimens after conventional FSW fractured mostly in the HAZ region on the AS according to 45° shear planes, while the rupture of the high-speed FSW specimens occurred in the NZ, as shown in Figure 8. The fracture location of the conventional FSW joint was coincident with the region of lower microhardness value, while the rupture of the HSFSW specimens could indicate the influence of the structural characteristics in NZ. In order to understand the failure reasons, the analysis of the fracture tensile surfaces was performed.

### 3.4. Fractography

The SEM observation of the fractured surfaces of the tensile specimens revealed different fracture behavior of the welds after conventional and high-speed FSW. Figure 9a presents SEM micrograph showing the fracture morphologies of the conventional FSW joint. The fracture surface mostly consisted of a large number of shallow and uniform dimples indicating predominantly a ductile behavior of the material. The formation of the dimples was initiated by the second phase particles by nucleating voids and micropores under increasing load. These particles could be observed inside the dimples on the fracture surface. The results of the EDX analysis (Spectrum 4) have confirmed that the Al-Fe-Mn-Si intermetallic particles acted as dimple nucleation sites, Table 2. The uniform distribution of relatively small particles provided the formation of equiaxed dimples in equivalent size, while the large particles up to 5 µm resulted in the discontinuous fracture of the dimples and formation of the quasi-cleavage facets that can reduce the ductility of the material. Nevertheless, the formation of dimples with different sizes demonstrated the ductile behavior of the joint before failure.

The morphology of the fracture surface of the HSFSW specimens was characterized by mixed ductile and cleavage fracture mode. The boundary between them can be seen along the weld center, Figure 9b. The above part presented the ductile fracture region with the dimples of a bigger size up to 25 µm compared to the conventional FSW specimen, Figure 9c. The fracture starts from the breakages of particles, which initiate the formation of microvoids at the grain boundaries. Large irregular particles of the intermetallics or their conglomerations, which were found at the bottom of the wide dimples lead to the discontinuous fracture of the dimples. These observations are attributable to the lower plastic deformation during the tensile test compared to the conventional FSW joint. The bottom part exhibited the morphology of the cleavage fracture represented the coarse cleavage facets which were elongated in a certain direction, presumably in the direction of crack propagation and separated by tear ridges, as presented in Figure 9d. Although small dimples could be observed at the edge of the facets, the domination of the cleavage facets indicated the brittleness mode, resulting in a very quick failure of the specimens.

## 4. Discussion

### 4.1. Microstructure Evolution and Microhardness Distribution

It was revealed that the microstructure in NZ had undergone changes during the HSFSW compared to the conventional FSW process. The decrease in grain size by an increase in the welding speed could be explained by the combination of many factors. On the one hand, lower linear energy supplied in the high-speed FSW (Table 1) resulted in shorter cooling cycle that inhibited grain growth. Longer dwell time at elevated temperature on the material in the NZ of the conventional weld caused an increase in the size of some nucleated grains, resulting in the formation of an inhomogeneous larger grain microstructure. In addition, the results obtained by numerical simulation have presented that the strain rate by the HSFSW was significantly higher compared to the FSW by lower welding speed. As Figure 10a compares, the strain rate in the cross-sections of the studied welds behind the probe in the NZ was more than two times higher in the HSFSW joint in comparison with conventional FSW. Supposedly, a higher strain rate promoted grain nucleation during dynamic recrystallization that could be a reason for the uniform grain distribution in the NZ microstructure.

Considering the distribution of the microhardness across the studied welds, a significant difference in microhardness values in the NZ of the conventional and HSFSW joints has not been found, but the hardness in the softened region outside NZ was significantly improved by a high-speed FSW process. As mentioned in the literature review, the microhardness distribution through the friction stir welded Al-Mg-Si joint can be explained by the processes of dissolution, coarsening, and reprecipitation of the strengthening precipitates depending on the temperature distribution in the different weld regions. Figure 11 presents the temperature fields of the studied joints on the upper surfaces, which were calculated using the CFD based numerical model. The most interesting aspect is that the temperature fields in the HSFSW joint were significantly narrower and shifted to the AS.

The same behavior of the temperature distribution was observed at the distance of 1 mm from the upper surface where the microhardness was measured. The temperature profile versus distance from the weld center was built on the cross line immediately behind the probe, Figure 10b. Since the maximum temperature was almost the same for the studied joints and reached 500 °C, it can be concluded that the strengthening precipitates were completely dissolved in the NZ of both welds. It led to a hardness fall in this zone of both welds; and the hardness level was approximately the same for both parameters sets. However, a small increase in microhardness of the HSFSW joint can be attributed to grain refinement.

Precipitation dissolution and coarsening in the HAZ, which represents the weakest region of the FSW joint, is responsible for a decrease in hardness in the HAZ compared to the base metal. Since these processes are affected by temperature, the correlation between hardness and temperature distributions across welds can be revealed. As can be concluded from Figure 11, the temperature gradient was larger by HSFSW, which resulted in the rapid cooling rate. Shorter temperature impact prevented the coarsening of the strengthening precipitates due to the reduction of the exposure time at elevated temperatures. Moreover, the lower temperature on the RS of the HSFSW joint in the area of a hardness decrease may cause of the partial dissolution of the strengthening precipitates compared to the conventional joint. Therefore, as a result of different temperature histories during welding, the decrease of the hardness level in the softened region of the joint welded at higher welding speed was not as significant as in the case of the conventional weld. The shift in the microhardness drop to the AS in the case of the HSFSW was also related to the distribution of temperature fields shifted to the AS, Figure 11a. The higher temperature affecting the AS caused an overaging resulting in the significant hardness drop on the AS compared to the hardness level on the RS.

### 4.2. Influence of Welding Speed on the Tensile Properties 

The existing literature on features of the ROL formation in the friction stir welds is extensive and focuses particularly on the correlation between heat input and ROL configuration as well as its influence on the mechanical characteristics of the joint [44,45,46,47,48]. A number of studies have claimed that the formation of a discontinuous oxide lines in the weld has slight influence on the mechanical properties of the as-welded joint. Controversy, a presence of a continuous oxide zigzag line deteriorates the weld property, since the fracture appears through the oxide line. However, previous research findings of the correlation between heat input, which is determined by the welding parameters, and ROL configuration was contradictory. Several reports have shown that the higher heat input (at high rotation speeds or low welding speeds) promoted the tool stirring process and thereby contributed to the eliminating of the negative effect of the ROL. On the other hand, different findings were obtained by [44].

In the current study, the ROL possessed a continuous zigzag configuration in both studied FSW joints, but the mechanical properties of the joint welded by lower welding speed were not affected by the oxide line, since the weld fracture occurred in the HAZ region. The tensile properties were affected by hardness distribution, as the lower hardness value was detected in the HAZ. The fracture surface of the produced conventional FSW joint represented the ductile fracture pattern with a large number of dimples that indicated the ductile behavior of the joint. As presented in Figure 8, the fracture path of the HSFSW specimens was coincident with the trace of the zigzag line, indicating that the ROL was a reason for the joint rupture. These results suggest that the ROL in the joint welded with higher heat input (HSFSW) had a detrimental effect on the mechanical properties while it did not influence on the property of the conventional FSW joints produced with lower heat input. It can be assumed that by increasing the welding speed the stirring process was insufficient due to the higher weld pitch (Table 1) and higher strain rate (Figure 10a), meaning a movement of the material by the tool was too fast. In such condition a good bonding between material layers was not achieved, which also confirmed the formation of cracks along with the oxide zigzag line in the high-speed weld.

## 5. Conclusions

An Al-Mg-Si aluminum alloy was friction-stir-welded at different welding speeds in order to determine the influence of the high welding speed on the joint properties. The results obtained in the present study were drawn as follows:(1)Increasing welding speed resulted in the formation of the finer and uniformly distributed grain microstructure in the nugget zone (NZ) as well as severely deformed microstructure in the transition zone;(2)The high-speed friction stir welding was an effective way to increase an overall hardness level in the softened region of the joint;(3)The conventional FSW joint fractured in the place where the hardness level was the lowest, while the tensile properties of the high-speed weld were affected by the formation of the thick oxide lines in NZ;(4)The fracture surface of the conventional FSW joint indicated ductile behavior of the material, while the HSFSW specimens were characterized by mixed ductile and cleavage fracture mode;(5)The results of the numerical simulation have shown that by high-speed welding the temperature gradient was larger and the temperature fields were shifted to the advancing side. The strain rate calculated using the CFD method in the cross-sections behind the probe in NZ was more than two times higher in the HSFSW joint in comparison with the conventional FSW joint.

## Figures and Tables

**Figure 1 materials-12-04211-f001:**
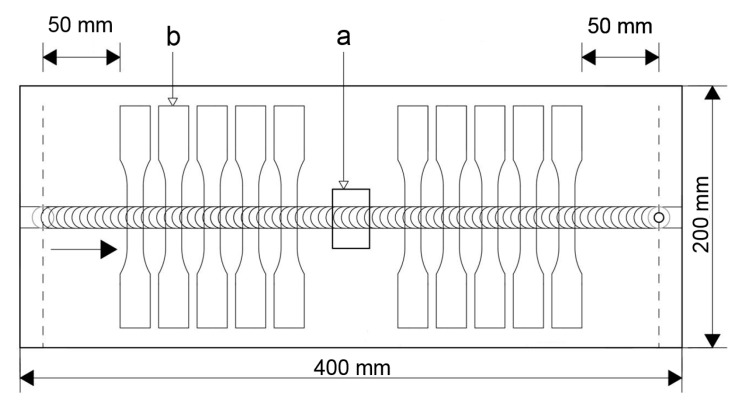
Locations of the specimens (a) for optical microscopy and the hardness measurement and (b) for the tensile test.

**Figure 2 materials-12-04211-f002:**
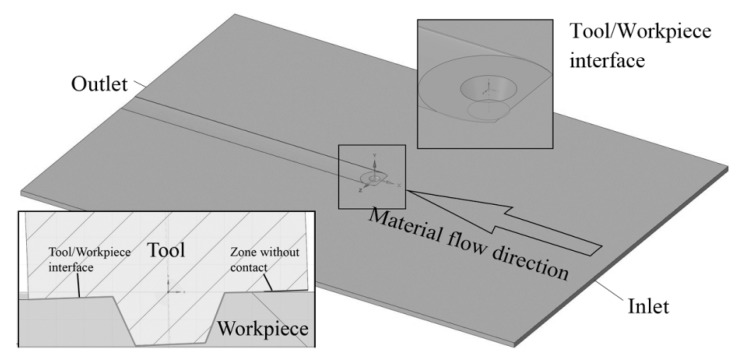
Scheme of the domain, tool position, and tool geometry.

**Figure 3 materials-12-04211-f003:**
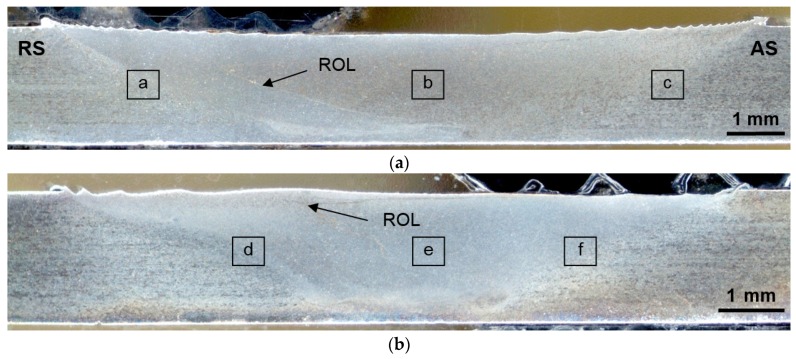
Transverse metallographic section of: (**a**) conventional friction stir welding (FSW) and (**b**) high-speed FSW (HSFSW) joint representing zigzag remnant oxide line (ROL).

**Figure 4 materials-12-04211-f004:**
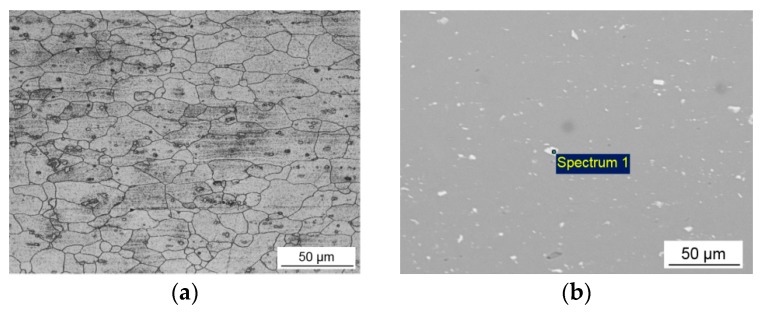
Microstructure of the base metal: (**a**) optical microscopy and (**b**) SEM micrograph.

**Figure 5 materials-12-04211-f005:**
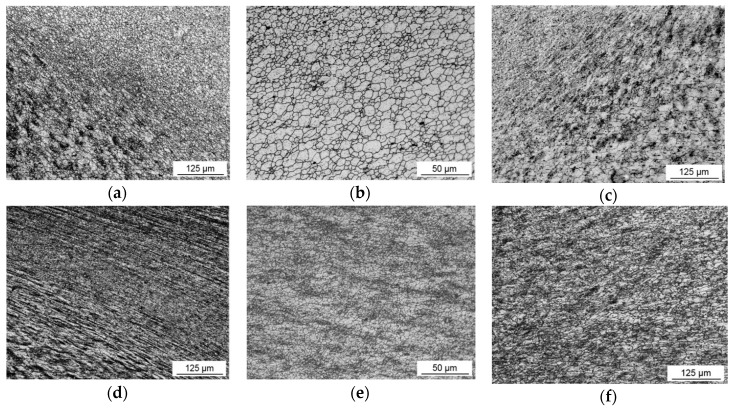
The microstructure in the different zones of the conventional FSW joint: (**a**) thermo-mechanically affected zone (TMAZ; retreating side (RS)); (**b**) nugget zone (NZ); (**c**) TMAZ (advancing side (AS)) and HSFSW joint: (**d**) TMAZ (RS); (**e**) NZ; and (**f**) TMAZ (AS).

**Figure 6 materials-12-04211-f006:**
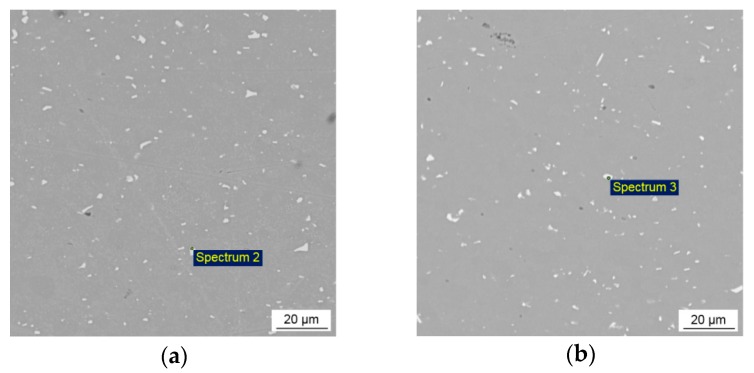
SEM images demonstrating the morphology and distribution of the second particles in the NZ of: (**a**) conventional FSW and (**b**) HSFSW joints.

**Figure 7 materials-12-04211-f007:**
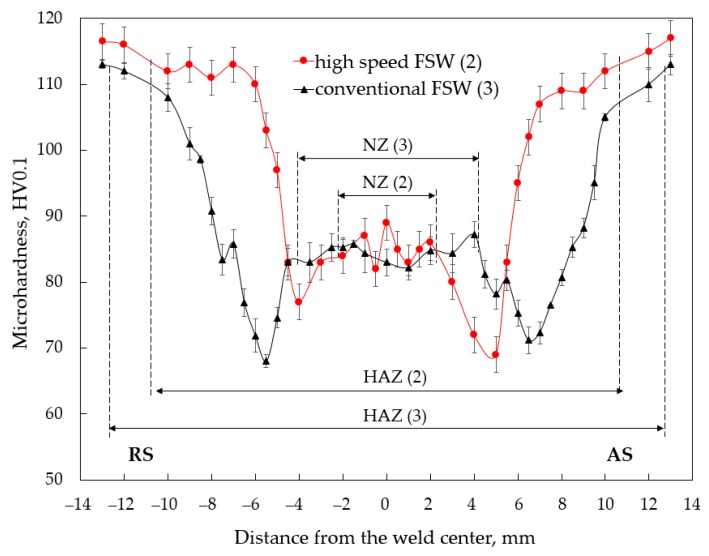
Microhardness profile of the studied joints: (2) high speed FSW and (3) conventional FSW.

**Figure 8 materials-12-04211-f008:**
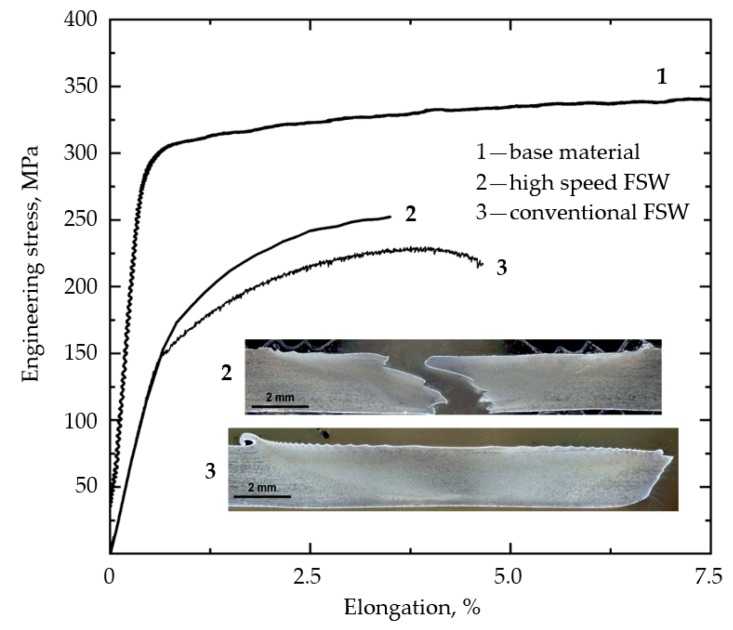
Tensile properties and fracture locations of the studied joints.

**Figure 9 materials-12-04211-f009:**
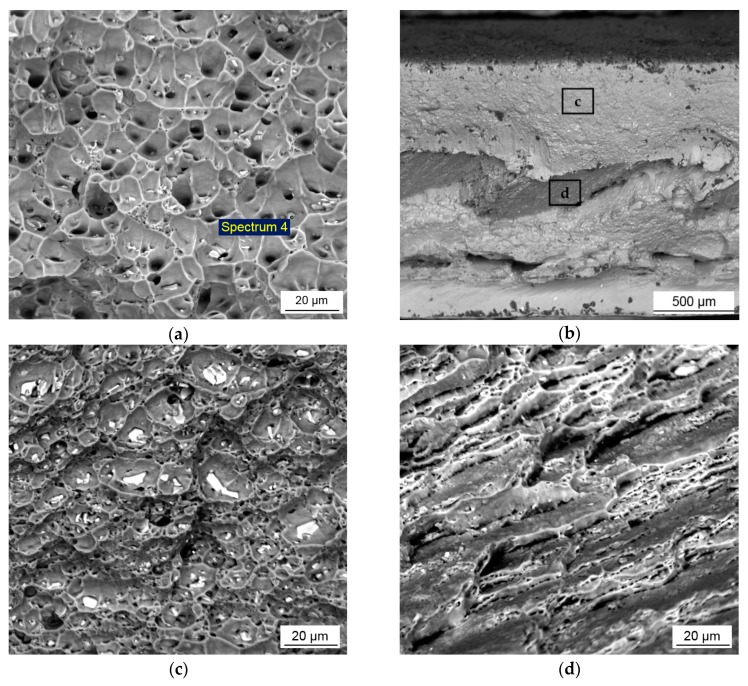
Scanning electron fractographs of tensile specimens: (**a**) fracture surface of the conventional FSW joint, showing dimple ductile fracture with the intermetallic particles inside dimples; (**b**) overview of the fracture surface of the HSFSW joint; (**c**) dimple fracture; and (**d**) cleavage fracture mode.

**Figure 10 materials-12-04211-f010:**
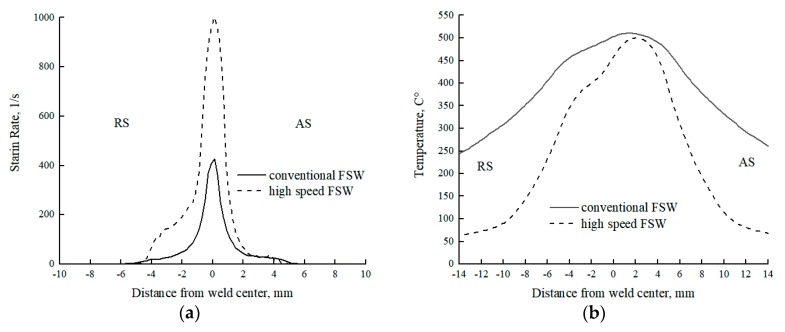
Calculated (**a**) strain rate profile and (**b**) temperature profile at the distance of 1 mm from the upper surface behind the probe.

**Figure 11 materials-12-04211-f011:**
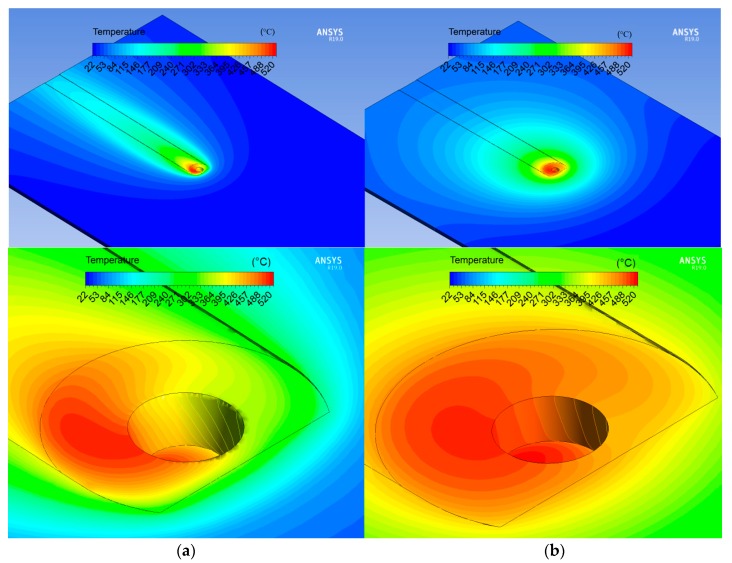
Temperature fields at the upper surfaces: (**a**) high-speed FSW and (**b**) conventional FSW.

**Table 1 materials-12-04211-t001:** Weld matrix.

FSW Type	v, mm/min	N, rpm	F_z_, kN	α	WP, mm/rot	Q_total_, Watt	l, J/mm
Conventional	200	710	4	2°	0.28	523	157
High-speed	2500	2100	6	1.2°	1.19	2222	53

**Table 2 materials-12-04211-t002:** Chemical compositions of the intermetallic inclusions (wt%).

Zone	Spectrum	Al	Fe	Mn	Si	Cr	Cu	Ni
BM	1	68.45	17.70	6.21	6.55	0.19	0.66	0.24
NZ-FSW	2	62.59	21.62	7.19	7.74	0.17	0.69	-
NZ-HSFSW	3	65.39	19.97	6.68	7.06	0.19	0.58	0.12
Fracture-FSW	4	62.09	22.36	6.85	7.82	0.19	0.69	-

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
