# Peer review of "Metallurgical and Mechanical Characterization of High-Speed Friction Stir Welded AA 6082-T6 Aluminum Alloy"

_materials, 2019, doi:10.3390/ma12244211_

Round 1

Reviewer 1 Report

Authors need to improve the english and grammar particularly at the introduction section.

Authors only discuss about one process parameter for HSFSW. is this the optimized one? If so what is the basis?

Microhardness profile shows that the NZ2 zone hardness for HSFSW and conventional FSW are almost similar. Rather, NZ3 zone microhardness reduced for HSFSW compare to conventional FSW. But, Fig.7 shows that HSFSW has hisher tensile strength compare to conventional one. Need a proper explanation.

Authors claimed that the grain size reduced during HSFSW. if so then both hardness and toughness/elongation should be increased. But elongation for HSFSW also decreased compare to conventional one.

Figure 5 clearly shows a huge difference in grain size. It will be better to explain with some definite value of grain size.

The peak temperature calculated by CFD seems to be much higher compare to the real one. It is always better to compare the numerical data with the experimental one. 

In point 2 in conclusion part: authors mentioned "overall hardness level in the softening region of the joint". Need clear explanation about the softening region.

In point 3 in conclusion part:"thick oxide lines in NZ due to a lack of bonding during the stirring process". "lack of bonding" is a very confusing word. Authors need to be explain it properly.

In point 5 in conclusion part: It is very hard to conclude anything only on the basis of CFD results about peak temperature atleast.

Author Response

Thank you for the remarks. The comments on them are presented below:

1. Authors need to improve the english and grammar particularly at the introduction section.

The grammar was brushed up.

2. Authors only discuss about one process parameter for HSFSW. is this the optimized one? If so what is the basis?

The process parameters were selected based on the bibliographic references as well as results of the previous research. The information about the parameters selection was added; it was described in detail in Section 2.1. Friction stir welding.

3. Microhardness profile shows that the NZ2 zone hardness for HSFSW and conventional FSW are almost similar. Rather, NZ3 zone microhardness reduced for HSFSW compare to conventional FSW. But, Fig.7 shows that HSFSW has higher tensile strength compare to conventional one. Need a proper explanation.

Figure 6 (Figure 7 in the new version) has been modified in order to improve the visibility of the presented microhardness profiles. The average microhardness in the NZ of the HSFSW joint (NZ(2)) was slightly higher and inhomogeneous compared to the conventional weld, and it was found to be around 85 HV (From text – 3.3. Microhardness and tensile behaviour). The tensile strength was also higher for the HSFSW joint compared to the conventional weld (Figure 8 in the new version). Therefore, the microhardness results corresponded to the results of the tensile tests.

4. Authors claimed that the grain size reduced during HSFSW. if so then both hardness and toughness/elongation should be increased. But elongation for HSFSW also decreased compare to conventional one.

The grain size in the NZ was reduced by HSFSW process (described in Section 3.2. Microstructural characteristics). The microhardness in the NZ was slightly higher in case of HFSW due to the reduction in grain size. The decrease in elongation for the HSFSW joint was caused by the influence of the remnant oxide line (ROL) formed in the nugget zone. The influence of the ROL was described in Section 4.2. Influence of welding speed on the tensile properties.

5. Figure 5 clearly shows a huge difference in grain size. It will be better to explain with some definite value of grain size.

The information about grain size in the nugget zones of the studied welds was expanded (Section 3.2. Microstructural characteristics). The grains in the NZ of the conventional FSW joint varied in the wide size range from 1 to 16 µm; the average grain size was around 6,1 µm. The average size of the equiaxed grains in the NZ of the high speed weld was about 3.8 µm.

6. The peak temperature calculated by CFD seems to be much higher compare to the real one. It is always better to compare the numerical data with the experimental one. 

In the present work, the peak temperature calculated by CFD was about 500°C for both studied welds. Since the temperature in the weld center is difficult to measure, the thermal history during FSW has been normally provided using thermal simulation. Despite the difficulties in the experimental measurement, e.g. the highest temperature which was provided using the thermocouples located in the friction stir welding tool was 527°C [1]. The results of the thermal simulation [2-5] have shown that the pic temperature of about 500°C, depending on the welding parameters and welded alloy, is typical for FSW of aluminium alloys.

In our previous research [6], the experimental peak temperature measured on the upper surface of the welded AA6082-Т6 sheets at the distance of 1 mm from the weld center was 450°C. That allows to suppose that the peak temperature in the weld center should be about 500°C.

Gerlich, A. Stir Zone Microstructure and Strain Rate during Al 7075-T6 Friction Stir Spot Welding. Metall. Mater. Trans. A. 2006, 37A, 2773-2786. Jata, K.V. et al. Continuous dynamic recrystallization during friction stir welding of high strength aluminum alloys. Scr. Mater. 2000, 43, 743–749. Sato, Y.S. et al. Microstructural Evolution of 6063 Aluminum during Friction-Stir Welding. Metall. Mater. Trans. A. 1999, 30A, 2439-2437. Khandkar, M.Z.H. et al. Prediction of temperature distribution and thermal history during friction stir welding: input torque based model. Sci. Tech. Weld. Join. 2003, 8(3), 165-174. Yang, B. et al. Banded microstructure in AA2024-T351 and AA2524-T351 aluminum friction stir welds Part I. Metallurgical studies. Mater. Sci. Eng. A, 2004, 364, 55–65. Golubev, I. et al. Developing finite element model of the friction stir welding for temperature calculation. 2014 METAL 2014 - 23rd International Conference on Metallurgy and Materials, Conference Proceedings, 1242-1248.

7. In point 2 in conclusion part: authors mentioned "overall hardness        level in the softening regionof the joint". Need clear explanation      about the softening region.

We’re grateful for such an important remark. In order to avoid misunderstanding, the softening region was renamed into softened region as this term is common. The meaning of the term “softened region” was added in Section 1. Introduction – “Numerous FSW studies of the 6xxx series alloys [9-15] have shown that the processes of dissolution and growth of the strengthening precipitates in the nugget zone (NZ), thermo-mechanically affected zone (TMAZ), and heat-affected zone (HAZ) lead to the softening in these zones, which together formed the softened region”. The term “overall softening” referred to the hardness level in the softened region.

8. In point 3 in conclusion part:"thick oxide lines in NZ due to a lack of bondingduring the stirring process". "lack of bonding" is a very confusing word. Authors need to be explain it properly.

Point 2 in Conclusions: the term “lack of bonding” was deleted, since the tensile properties of the high-speed weld were affected by the formation of the thick oxide lines in NZ”.

Under “lack of bonding” it was meant the following: “It can be assumed that by increasing the welding speed the stirring process was insufficient due to the higher weld pitch (Table 1) and higher strain rate (Figure 10a), meaning a movement of the material by the tool was too fast. In such condition a good bonding between material layers was not achieved”. The use of “lack of bonding” was avoided.

9. In point 5 in conclusion part: It is very hard to conclude anything only on the basis of CFD results about peak temperature at least.

Unfortunately, since the experimental measurement of the temperature in the weld center is very difficult, the peak temperature could be determined only using CFD model. According to our previous research [6], the experimental peak temperature measured on the upper surface of the welded AA6082-Т6 sheets at the distance of 1 mm from the weld center was 450°C. That allows to suppose that the peak temperature in the weld center should be about 500°C.

Reviewer 2 Report

What are the average 2nd phase particle sizes within the NZ for the conventional FSW and HSFSW joints? Can the authors show suitable SEM images to show the particles morphology and distribution in the NZ? Referring to the microstructures in Fig. 5 and the micro-hardness profiles of the Fig. 6, the average grain size in NZ is significantly reduced for the FSW joints, but the micro-hardness of FSW joints is also lower than that of BM in the NZ. It seems against the definition of Hall-Petch relation. What are the metallurgical factors for the reduction of Hv with the refining of grain sizes in the NZ of both FSW joints?

Author Response

Thank you for the remarks. The comments on them are presented below:

1. What are the average 2nd phase particle sizes within the NZ for the conventional FSW and HSFSW joints? Can the authors show suitable SEM images to show the particles morphology and distribution in the NZ?

The SEM images showing the morphology and distribution of the second phase particles in the NZ of the conventional and high speed welds have been provided in Figure 6. The average size of the particles in both welds was measured and this information was added to the text. The average size measured on a large number of the particles in the NZ was about 2.5 and 2 µm for the conventional and HSFSW joints, respectively. A slight decrease in particle size in the NZ of both welds in comparison with BM can be explained by the fragmentation of the second phase particles by the stirring process.  

2. Referring to the microstructures in Fig. 5 and the micro-hardness profiles of the Fig. 6, the average grain size in NZ is significantly reduced for the FSW joints, but the micro-hardness of FSW joints is also lower than that of BM in the NZ. It seems against the definition of Hall-Petch relation. What are the metallurgical factors for the reduction of Hv with the refining of grain sizes in the NZ of both FSW joints? 

The average grain size in the NZ was reduced by HFSW, as can be concluded from the comparison of Figure 5b with Figure 5e as well as from the results of the grain size measurements (6 µm for FSW vs 3.8 µm for HSFSW). The microhardness in the NZ was slightly higher for the HSFSW joint compared to the conventional one, as shown in Figure 7 (Figure 7 in the new version has been modified in order to improve the visibility of the presented microhardness profiles). Therefore, the increase in microhardness in the NZ of the HSFSW joint can be explained with decrease in the grain size in the NZ.

Reviewer 3 Report

In this manuscript, Naumov et al. investigated the effect of the HSFSW on the mechanical properties and their relations to microstructural characteristics of butt FSW joints with the use of 6082-T6 aluminum alloy. This is an interesting work that is suitable to publish in Materials. There are some concerns to be addressed:

For tensile samples, authors should cut samples from two directions (refer Fig. 1 in this paper (Microstructure, mechanical properties and corrosion of friction stir welded 6061 Aluminum Alloy(https://arxiv.org/pdf/1511.05507.pdf))). In Fig. 8a, authors should provide more spectrum to see difference among edge, white matter, and hole. Please provide error bar for each data point of microhardness. For references, please cite more recent articles. Please edit and polish your writing.

Round 2

Reviewer 1 Report

Accepted in its present form.